# Effect of Surface Treatment of Polypropylene (PP) Fiber on the Sulfate Corrosion Resistance of Cement Mortar

**DOI:** 10.3390/ma14133690

**Published:** 2021-07-01

**Authors:** Yanyan Hu, Linlin Ma

**Affiliations:** 1Department of Materials Science and Engineering, Xi’an University of Architecture and Technology, Xi’an 710055, China; mll2336@126.com; 2Key Laboratory of Green Building in Western China, Xi’an University of Architecture and Technology, Xi’an 710055, China

**Keywords:** polypropylene fiber, surface modification, cement mortar, resistance to sulfate attack

## Abstract

Sulfate erosion is one of the most complex and harmful chemical corrosion actions. Following sulfate erosion, concrete expands, cracks, dissolves, peels off, and decreases in strength, which affects the durability of structures. Polypropylene fiber (PP) is widely used in various concrete structures because of its good mechanical properties and chemical corrosion resistance. However, PP fiber has a number of shortcomings, such as a smooth surface, poor hydrophilicity, lack of active groups in the molecular chain, and agglomeration and poor dispersion in cement-based materials. These issues limit its application in cement-based materials. Although the use of a silane coupling agent to modify the surface of PP fiber is effective, the influence of treated PP fiber on the sulfate resistance of cement-based materials is not significant. In this study, a PP fiber treated with a silane coupling agent was used to examine effects of different cement-to-sand ratios (C/S) and dosages of the treated PP fiber on the sulfate erosion resistance of cement mortar. Furthermore, the apparent morphology, mass loss rate, flexural strength, corrosion resistance coefficient, and microstructure of the concrete were investigated by X-ray diffraction (XRD) and scanning electron microscopy (SEM). The results revealed that the PP fiber became rough after modification. Active groups were introduced on the fiber surface, which were well dispersed in the mortar and formed a good network distribution structure in the cement mortar, thereby slowing the erosion rate of the PP fiber mortar in a sodium sulfate solution. At a C/S ratio of 1:1 and a treated fiber dosage of 0.6%, the treated fiber mortar has exhibited good sulfate resistance. In addition, the monofilament fiber immersion test revealed that a layer of sodium sulfate crystals was deposited on the fiber surface, thereby increasing the roughness of the fiber surface and the pull-out force of the fiber from the cement matrix, this result indicated that the interfacial adhesion between the treated PP fiber and cement matrix was improved, which in turn led to the improvement in the sulfate erosion resistance of the treated PP fiber.

## 1. Introduction

Concrete is a widely used building material. However, due to its limitations, including poor tensile properties and poor crack resistance, which directly affect its durability, its use in engineering applications is limited. Sulfate erosion is one of the most complex and harmful chemical corrosion actions. Following sulfate erosion, concrete expands, cracks, dissolves, peels off, and decreases in strength, which affects the durability of structures. Fiber is a new type of an environmentally friendly and green high-performance material, and can be used in cement materials to improve bending, impact properties, toughness, and corrosion resistance [1,2,3]. In addition, due to the bridging effect of fibers, resistance to crack propagation increases [4,5,6].

Recently, synthetic fibers have become more valuable as reinforcements for cementitious materials [7], and to serve as an alternative to asbestos, steel, and glass fibers due to health and economic reasons [8,9,10]. Sarvaranta [11] and Mikkola [12] reported that, among different fibers, polypropylene (PP) fiber-reinforced concrete is the most effective. PP fiber exhibits several advantages, including cost effectiveness, ductility, corrosion resistance, good thermal stability (high melting point), and high stability in the alkaline environment of concrete [12]. Mustea et al. [13] believe that the dosage of 1.0 kg PP fibers per 1 m^3^ mortar represents an optimal value for plaster mortars. Guo et al. [14] showed that a suitable amount of PP fiber is 1.3 kg/m^−3^. Under this dosage, PP fiber can not only improve the construction performance of aeolian sand mortar, but also improve its basic mechanical properties. Chajec et al. [15] reported that the addition of PP fiber to the concrete mixture does not increase the compressive and tensile strength of concrete. Ye et al. [16] reported that adding 1.5% PP fiber can improve the flexural strength of concrete.

In addition, due to the hydrophobic surface of the polymer (i.e., it does not absorb water), it does not interfere in the hydration reaction of concrete [16]. However, the PP chain structure is chemically inert and hydrophobic, and has a low surface energy. When PP fibers are mixed into the concrete mixture, the fibers form clusters, and uniform distribution cannot be achieved. Clusters of fibers often trap a considerable amount of air, adversely affecting the mechanical properties of fiber-reinforced concrete [17]. Therefore, to extend the application range of the PP fiber concrete, the fiber surface needs to be treated via the increase in the surface roughness of the polymer and introduction of polar groups on its surface, thereby improving the dispersion of the fiber in the concrete [18,19]. At present, many surface treatment methods exist for PP fibers, such as plasma treatment, surface oxidation and etching, photooxidation surface treatment, and radiation grafting treatment [20,21,22,23,24]. López-Buendía [20] and Akand et al. [21] reported that the surface chemical properties and morphology of chemically treated PP fibers are improved, and the adhesion performance between the fibers and concrete, and the fibers’ surface roughness, is increased. The mechanical properties of treated fiber-reinforced concrete composite materials are improved in comparison with those of untreated fiber concrete. Pietro [22] and Zhang et al. [23] reported that PP fiber modification can significantly improve the (a) uniformity of fiber dispersion in the cement mortar matrix, (b) the adhesion between the fiber and concrete matrix, and (c) the toughness and cracking of the concrete. Denes [24] and Li et al. [25] found that treated PP fiber can improve the mechanical properties of cement-based composites (viz. flexural strength and flexural toughness). Gang [26] and Dai et al. [27,28] reported that the incorporation of treated PP fibers can improve the freeze-thaw resistance, impermeability, and erosion resistance of concrete. The treated fiber has good adhesion to concrete and is evenly dispersed in the concrete. It can prevent cracks, strengthen and toughen the concrete, and improve the interface performance between the fiber and the concrete.

However, these methods have many shortcomings, such as surface oxidation; etching methods used to improve the adhesion reduce fiber strength; the chemical reagents used are very harmful to the environment; and the equipment continuity degree and stability of plasma surface treatment are poor. In addition, the investment required for the equipment used for photooxidation surface and radiation grafting treatments is very high, the optimum process conditions are challenging, and industrial production is difficult. In this paper, nano-sized calcium carbonate and a silane coupling agent were used to modify PP fiber.

Studies of treated PP-fiber-reinforced concrete have mainly focused on crack resistance, mechanical properties, and impermeability. However, the sulfate dry-wet cycle corrosion resistance of treated PP-fiber-reinforced cement composites remains unknown. Therefore, in this study, the effects of the treated PP fiber dosage and different cement-to-sand ratios (C/S) on the sulfate erosion of cement mortar were examined. The mechanism of sulfate erosion resistance of cement mortar with treated PP fiber is discussed with reference to the surface observation, mass loss rate, and corrosion resistance coefficient.

## 2. Materials and Methods

### 2.1. Raw Materials

The cement used for the experiments was ordinary Portland cement (OPC), with the strength grade of 42.5 and produced by Shanxi Jidong Cement Factory (Xi’An, Shanxi, China) was used. The chemical compositions of cement are listed in Table 1. The physical properties of this cement are presented in Table 2. Because a fine aggregate, i.e., natural river sand (Zhouzhi, China) with a fineness modulus of 2.7, was used in this experiment, a powdered naphthalene-based superplasticizer (Tongcheng Construction Technology Co., Ltd., Shanxi, China) was selected. The diameter and length of the PP fiber (Shandong Huimin Xinhang Chemical Fiber Products Co. Ltd., Binzhou City, China) were 50 μm and 19 mm, respectively. The main performance parameters of the PP fiber are presented in Table 3.

### 2.2. Surface Modification of the PP Fiber

First, the PP fiber was soaked in anhydrous ethanol (Tianjin Tianli Chemical Reagent Co. Ltd., Tianjin, China) for 12 h, washed, and dried to a constant weight. Second, deionized water, anhydrous ethanol, and the silane coupling agent (Nanjing Chuangshi Chemical Auxiliary Co. Ltd., Tianjin, China) were mixed together in a mass ratio of 1:9:40 to prepare the silane coupling agent solution by ultrasonication (JY92-2, Shanghai Jingxin Industrial Development Co., Ltd., Shanghai, China). First, a specific amount of the PP fiber was soaked in the silane coupling agent solution at 60 °C at room temperature for 7 h, followed by washing with deionized water and filtration, drying in a vacuum oven (Zhengzhou Changchengke Industry and Trade Co., Ltd., Zhengzhou, China) at 60 °C for 8 h, and cooling to room temperature. Next, the fiber was placed in a drying pan and dried to a constant weight.

Preparation of a nano-calcium carbonate dispersion solution with deionized water: First, water, sodium polyacrylate (Tianjin Cameo Chemical Reagent Co., Ltd., Tianjin, China), and nano-calcium carbonate (Chengdu McArch Chemical Co. Ltd., Chengdu, Sichuan, China) in a mass ratio of 400:1.2:7 was mixed, followed by stirring in a mixer at 80 °C for 1 h and weighing a certain amount of the silane coupling agent. The treated PP fiber was added to the prepared nano-calcium carbonate solution and stirred with a blender (Wuxi Xiyi Building Materials Instrument Co. Ltd., Wuxi, Jiangsu, China) at 80 °C for 2 h. After stirring was completed, the PP fiber was removed, washed with absolute ethanol, and dried to obtain the final treated PP.

### 2.3. Sample Preparation and Test Method

Preparation of fiber-reinforced cement mortar: First, sand, and cement were added in the mixer and stirred for 120 s, followed by the addition of the fiber for 120 s, and water and the water-reducing agent for 120 s. The preparation was completed and loaded into a 40 mm × 40 mm × 160 mm mold, followed by curing at room temperature for 24 h. After demolding, specimens were placed in a standard curing chamber at a temperature of 20 ± 3 °C and a relative humidity of greater than 90% for 7 days. A total of 150 sets of test samples was produced.

After samples were exposed to 5% Na_2_SO_4_ solutions (Tianjin Guangfu Science and Technology Development Co. Ltd., Tianjin, China) and cured for 30, 60, 90, 120, and 150 days, the samples were subjected to a sulfate erosion test using a dry-wet cycle immersion method. Next, the test block was soaked in a sodium sulfate solution for 1 day: it was soaked for 16 h and dried naturally for 8 h. The solution was changed every 15 days. After sulfate exposure, the specimens were removed and dried at 60 °C. The mass before and after exposure and the flexural strength (JJ-5, Wuxi Xiyi Building Materials Instrument Co. Ltd., Wuxi, Jiangsu, China) were measured. Table 4 lists the mortar mix design.

This test used the evaluation index of the mass loss rate and corrosion resistance coefficient of flexural strength, as shown in the following respective equations. According to the following equation of the Chinese National Standard GB/T 50082-2009 [29].
(1)$$\beta =\frac{M_{2}-M_{1}}{M_{1}}\times 100$$
where *β* is the mass loss rate of the specimen at a certain age (at two decimal places), *M*_1_ (kg) is the reference mass of the specimen before the test, and *M*_2_ (kg) is the mass of the specimen at a certain age.
(2)$$F_{n}=\frac{R_{2}}{R_{1}}\times 100$$
where $F_{n}$ represents the corrosion resistance coefficient of flexural strength (at two decimal places); *n* in $F_{n}$ represents the number of dry-wet cycles for sulfate corrosion resistance; *R*_1_ represents the flexural strength of the standard cured samples; and *R*_2_ represents the flexural strength of the samples under dry-wet cycles.

### 2.4. Analysis Techniques

Analysis was undertaken using Fourier transform infrared spectroscopy (FTIR, Bruker E-Q U IN O X55, Karlsruhe, Germany). The changes of functional groups on the surface of the fibers after treatment were tested. Before the determination, the fibers were dried in a vacuum oven at 40 °C for 2 h, and the dried fibers were cut into pieces with scissors and compressed (JYP-3, Tianjin Jiaxinhai Machinery Equipment Co. Ltd., Tianjin, China) into tablets with potassium bromide (Wilbo New Materials Co., Ltd., Jiangsu, China) for testing.

The wetting angle of the fibers (JY-82B Kruss DSA, Tokyo, Japan, Dataphysics OCA20) was measured as follows: A 5 cm long monofilament fiber was fixed on the support. After the water droplets were stabilized on the fiber surface for 15 s, the contact angle between the fiber and water was measured, and the average value was obtained by measuring 3 times. After the wet-dry cycle, the samples were subjected to microscopic analysis by X-ray diffraction (XRD, Rigaku D/Max 2400, Tokyo, Japan) and scanning electron microscopy (SEM, FEI Quanta 200, Denton, TX, USA).

## 3. Analyses and Discussions

### 3.1. Characterization of Untreated and Surface-Treated PP Fibers

Figure 1 shows the SEM images of the surface of treated and untreated PP fibers. As shown in Figure 1, the surface of the untreaded PP fiber is smooth, whereas the surface of the PP fiber treated by the nano-CaCO_3_-silane coupling agent is rough, and fine white nano CaCO_3_ particles are attached to the surface of the fiber.

Figure 2 shows the wetting angles of the untreated and treated PP fibers. The wetting angle of the untreated PP fiber was 137.2°. After treatment, the wetting angle decreased to 125.9° and the hydrophilicity increased by 8.9%, indicating that the untreated PP fiber exhibits lower wettability. This is because, following the composite modification of the silane coupling agent-nano calcium carbonate, some hydroxyl groups on the surface of the nano calcium carbonate react with sodium polyacrylate to produce hydrophilic carboxyl groups (COO–). Therefore, when in contact with water, the fibers exhibited good hydrophilicity. The FTIR spectrum (Figure 3) revealed that the treated PP fiber exhibits typical bands for CH_2_ stretching (2867.42 cm^−1^) and C=C bending (810.34 cm^−1^). The PP fiber surface exhibited C=O groups, which were observed as strong vibration peaks at 3390 and 1721.12 cm^−1^. The C=O group contributed to the adsorption of calcium carbonate molecules on the grafted PP fiber surface [30], indicating that the treated PP fiber surface contains active groups.

Figure 4 shows the dispersion of the untreated and treated PP fibers in water. Untreated PP fibers tended to accumulate and float unevenly in water. The treated PP fibers were uniformly scattered in water. Because calcium carbonate molecules can absorb water molecules, after hydrolysis, several hydroxyl groups were present on the calcium carbonate surface, which can improve the moisture absorption performance of the fiber. Therefore, the grafted and coated PP fibers can be uniformly dispersed in water [31].

### 3.2. Visual Observation

Figure 5 and Figure 6 show different C/S values and different dosages of treated PP fiber samples after erosion by repeating the sulfate dry-wet cycle 150 times.

Figure 5 shows that, with different C/S ratios and dosages of treated PP fiber, the pores on the surface of the test piece increase significantly. It can be seen from Figure 6 that after 150 cycles of sulfate, the edges and corners of the specimen were damaged and the surface was pitch-etched. In addition, with the increase in the number of wet-dry cycles, the accumulation of sodium sulfate on the surface of the specimen gradually increased. After a certain period of erosion, the concentration of sodium sulfate in the pores was higher than the concentration of the configured sodium sulfate solution, resulting in a concentration difference, causing damage to the specimen. With the increase in the dosage of fiber, the inside of the test block became fluffy, the pore structure was larger, and the corrosion solution could enter more easily. When the pore structure was loose, sulfate ions reacted with calcium hydroxide and hydrated calcium aluminate in the cement stone to form ettringite, causing volume expansion. When sulfate concentration was high, gypsum crystal precipitates and the volume of solid phase increased by 124%, leading to expansion and cracking of the cement mortar due to high expansion pressure in the cement slurry [31]. When the C/S radio was 1:1 and the dosage of fiber was 0.6%, the bond between the treated fiber and cement base was closer, and the intrusion of corrosive liquid was reduced. The integrity of the mortar sample was good.

### 3.3. Quality Loss

Figure 7 shows the mass loss rate of the treated PP fiber cement mortar at different C/S ratios and different dosages in a 5% sodium sulfate solution.

The same mass loss rate was observed for the specimens with different C/S ratios in the corrosion solution, and the mass of the specimens first increased and then decreased. After 120 wet-dry cycles of the sulfate erosion test, the quality of the cement mortar test blocks with 0.6 and 0.9% PP fiber content was improved, whereas after 150 cycles, the quality of all samples decreased (Figure 7a). After 120 cycles, the quality of samples with different mixing amounts decreased to some degree (Figure 7b,c). This is because, in the early stage of erosion, the specimen surface was dense, and the sulfate solution first reacted with the surface cement mortar. Sulfate adhered to the sample surface and increased the sample mass. With the progress of erosion, an increased amount of the sulfate solution entered the specimen and reacted with the hydrate of the internal cement slurry to form ettringite. The ettringite crystals precipitated, and the mortar quality was improved. With the increase in the erosion time, the amount of cement stone reacting with sulfate decreased, and the increase in the ettringite and gypsum formation led to the expansion and cracking of the specimen. Moreover, with the passage of time, sulfate invaded the interior of the sample, and sulfate crystals precipitated from the pores. As the cavity on the block surface further expanded, the quality of mortar started to decline, and surface pitting corrosion and fiber leakage were observed.

### 3.4. Corrosion Resistance Coefficient of Flexural Strength

Figure 8 shows the corrosion resistance coefficient of flexural strength of the treated PP fiber mortar test specimens under different C/S ratios.

With the increase in the number of wet-dry cycles, the corrosion resistance coefficient of flexural strength of the sample first increased and then decreased, and then tended to be stable (Figure 8). The corrosion resistance coefficients of flexural strength of the mortar specimens with the treated fiber were greater than those of the untreated fiber. In the Na_2_SO_4_ corrosion process, the corrosion resistance coefficients of flexural strength of the treated PP fiber samples with C/S ratios of 1:1, 1:1.5, and 1:2 were greater than that of a ratio of 1.0. At a C/S ratio of 1:2, the corrosion resistance coefficients of flexural strength of the treated PP fiber increased slowly and, after 60 cycles, it decreased more rapidly in comparison with that shown in Figure 8a,b. The lower the C/S ratio, the weaker the sulfate corrosion resistance of the fiber mortar. In addition, as can be observed from Figure 8, at a treated PP dosage of 0.6%, the corrosion resistance coefficients of flexural strength of the treated PP fiber were the highest, and at a treated PP fiber dosage of 1.5%, the corrosion resistance coefficients of flexural strength were the lowest.

With the increase in the treated PP fiber dosage, the surface area of the fiber increased, and the mortar could not wrap around the fiber, leading to the increase in the weak interface. When the weak interface was superior to the reinforcing effect of the fiber, the bending strength of the mortar decreased. In addition, the uniform dispersion of fibers in cement-based composites crucially affects their mechanical properties [32,33,34,35]. If the fibers are unevenly dispersed in cement-based materials, not only is the performance of fiber-reinforced cement-based composites affected, but stress concentration points are also formed, eventually serving as a weakness and a breakthrough point for the invasion of a corrosive medium, subsequently endangering the structural safety and reducing the service life of the structure. The treated PP fiber was uniformly dispersed in the cement mortar, thereby improving the sulfate corrosion resistance of the specimen.

### 3.5. Microstructural Investigation by XRD and SEM

Figure 9 shows the images of treated PP fiber soaked in a Na_2_SO_4_ solution for 7 h, 12 h, 3 days, and 7 days.

After immersing the treated PP fiber in a sodium sulfate solution for 12 h (Figure 9a) and 24 h (Figure 9b). After soaking for 3 days (Figure 9c), the fiber surface was covered by sulfate, and the fiber diameter started to increase. After the continuous immersion of the treated PP fiber in a Na_2_SO_4_ solution for 7 days, a layer of regular strip-like products adhered to the treated fiber surface, and the fiber diameter was further increased in comparison with that when it was immersed for 3 days (Figure 9d). The surface morphology of the treated PP fiber in the Na_2_SO_4_ solution changed. The treated PP fiber immersed in the Na_2_SO_4_ solution increased the fiber diameter, the fiber surface became rough, and the fiber could bond with the matrix more firmly (Figure 10).

Figure 11 shows the SEM images of PP fibers with different dosages after 150 cycles of dry-wet sulfate. Unreacted calcium hydroxide was observed in Figure 11b and c because the mortar was mixed with an appropriate amount of treated PP fibers to prevent the mortar from hardening before the formation of interconnected cracks and holes, effectively inhibiting the invasion of external SO_4_^2−^. Ettringite crystals were also observed. The expansion of ettringite possibly caused microcracks, but the growth of ettringite around the cracks, and the filling of micropores and microcracks helped to increase the matrix density. It can be seen from Figure 12 that the ettringite peak increased after sulfate attack. As can be observed in Figure 11a,d,e, ettringite crystals and calcium hydroxide were absent because the fibers were not mixed, and with an extremely high fiber dosage, the matrix was not dense, and connected pores were formed; hence, the etching liquid was able to enter more easily [36,37]. When the fiber content was 1.5%, a tiny sodium sulfate diffraction peak appears in Figure 12. Therefore, the larger dosage of the fiber and anhydrous Glauber’s salt with a higher sodium sulfate crystallization pressure produced a higher swelling stress on the pore wall, which also considerably accelerated sample damage.

## 4. Conclusions

To address the problems of the smooth surface of PP fibers, the poor hydrophilicity and poor dispersion of fibers in cement mortar, and inadequate bonding of the matrix, in this study the surface of PP fiber was treated with a silane coupling agent. The effect of treated PP fiber on the sulfate resistance of cement mortar was analyzed by visual observation, and via the resistance coefficient of flexural strength and microanalysis. The main conclusions in this paper are summarized as follows:The PP fiber surface treated with the silane coupling agent was successfully grafted with hydrophilic groups. The wetting angles of untreated PP fiber and treated PP fiber were 137.2° and 125.9°, respectively. The hydrophilicity of the treated PP fiber was increased by 8.9%.The corrosion resistance coefficients of flexural strength of cement mortar with the treated PP fiber were greater than those without PP. When the mortar ratio was 1:1 and the fiber content was 0.6%, the mass loss rate and flexural strength loss of the specimens were the lowest.The surface-treated PP fiber was immersed for different times in a Na_2_SO_4_ solution, and the change in the surface morphology and chemical composition were characterized by SEM and EDS. The surface roughness of the surface-treated PP fiber increased and the fiber diameter became larger. This improved the interface adhesion between the treated PP fiber and the cement matrix, which is the main reason for improving the sulfate erosion resistance of cement mortar.

## Figures and Tables

**Figure 1 materials-14-03690-f001:**
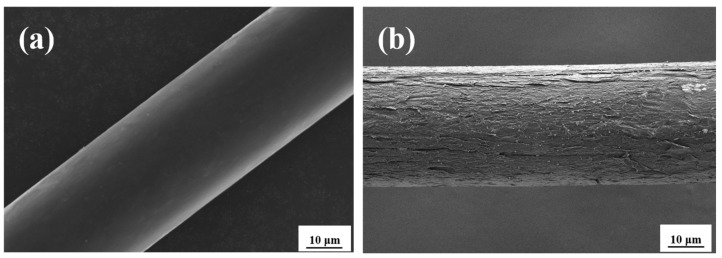
SEM image of PP fiber: (**a**) untreated; (**b**) treated.

**Figure 2 materials-14-03690-f002:**
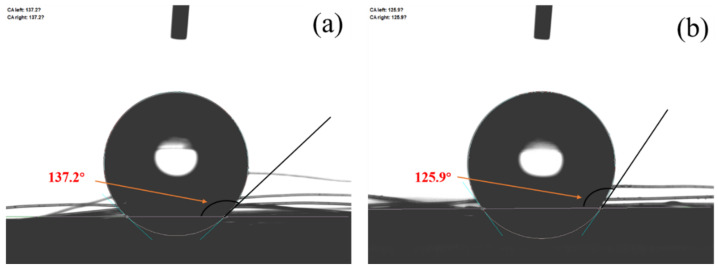
Wetting angle of PP fiber: (**a**) untreated PP fiber; (**b**) treated PP fiber.

**Figure 3 materials-14-03690-f003:**
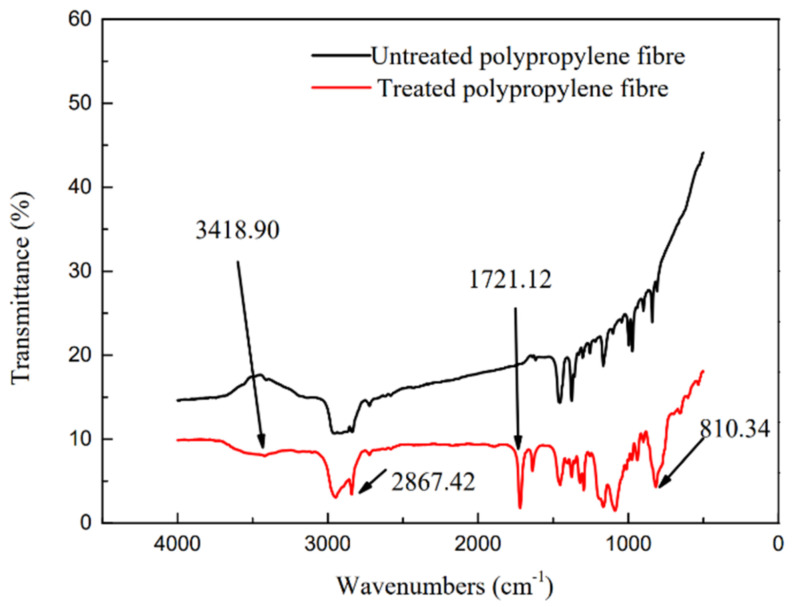
Infrared spectra of treated and untreated PP fibers.

**Figure 4 materials-14-03690-f004:**
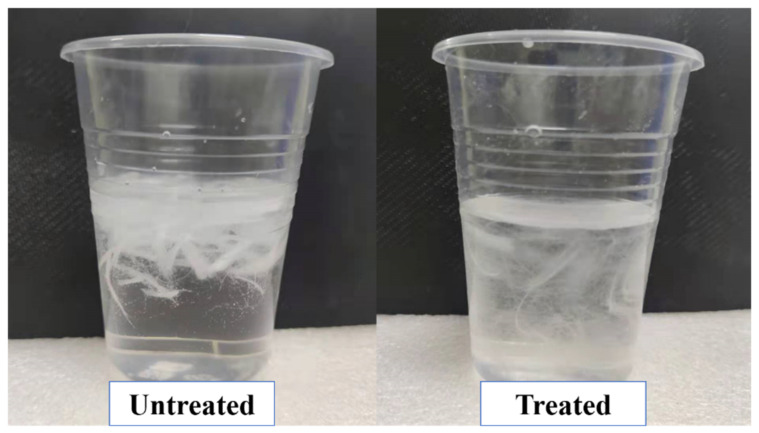
Dispersion of untreated and treated PP fiber in water.

**Figure 5 materials-14-03690-f005:**
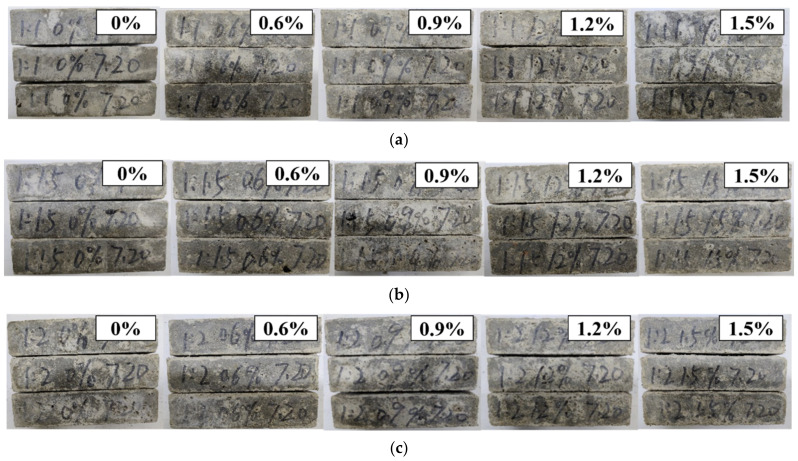
Appearance of PP fiber mortar specimens treated with different C/S values and different dosages after 150 dry-wet cycles: (**a**) 1:1; (**b**) 1:1.5; (**c**) 1:2.

**Figure 6 materials-14-03690-f006:**
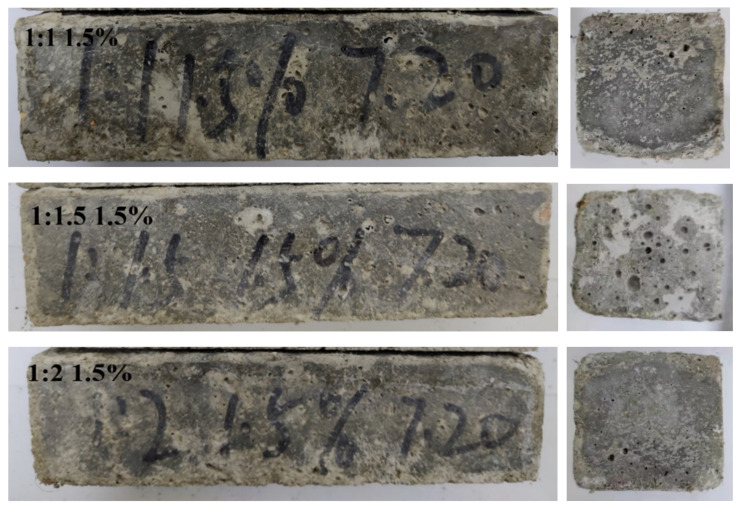
Appearance of C/S values of 1:1, 1:1.5, and 1:2, and 1.5% dosage of treated PP fiber mortar specimens after 150 dry-wet cycles.

**Figure 7 materials-14-03690-f007:**
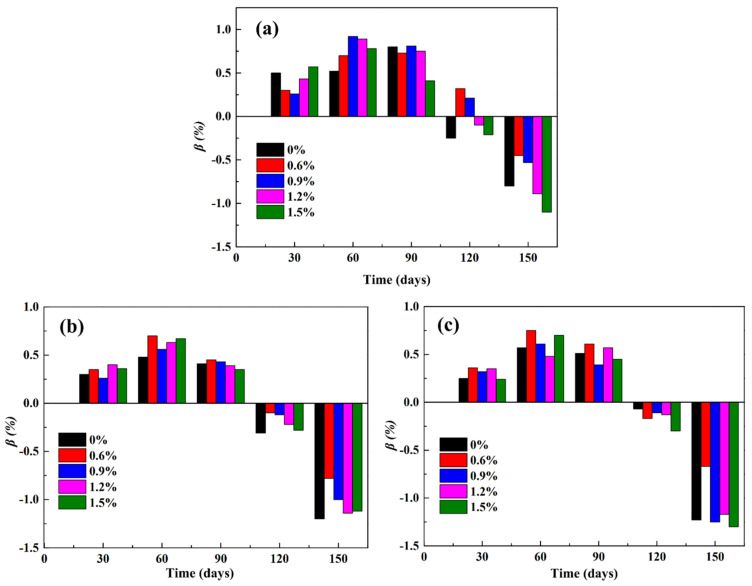
The mass loss rate of PP fiber mortar specimens treated with different C/S ratios: (**a**) 1:1; (**b**) 1:1.5; (**c**) 1:2.

**Figure 8 materials-14-03690-f008:**
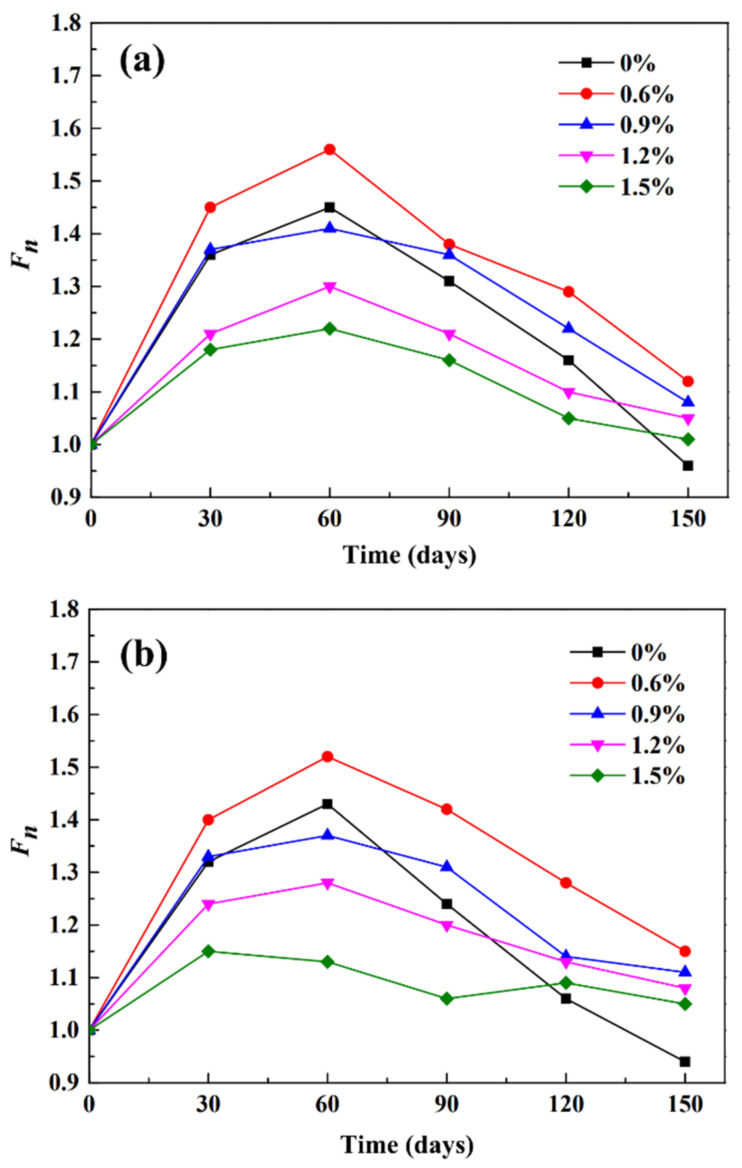
The corrosion resistance coefficient of flexural strength of the treated PP fiber mortar test specimens under different C/S ratios: (**a**) 1:1; (**b**) 1:1.5; (**c**) 1:2.

**Figure 9 materials-14-03690-f009:**
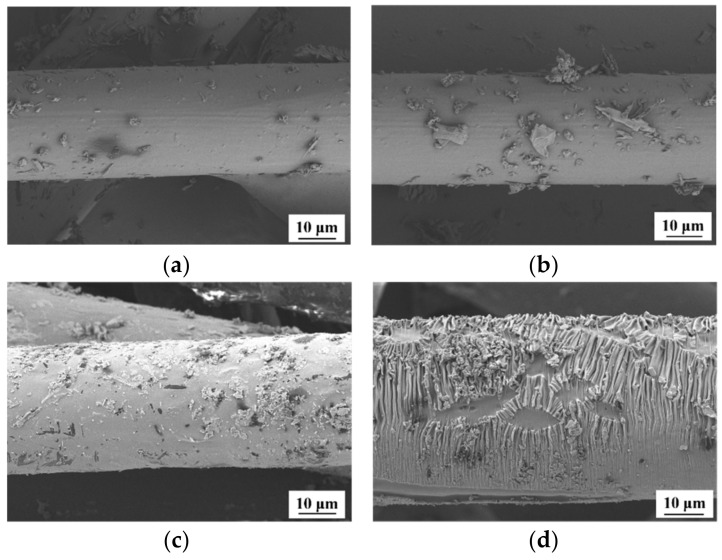
SEM of treated PP treatment fiber immersed in Na_2_SO_4_ solution: (**a**) 7 h; (**b**) 12 h; (**c**) 3 d; (**d**) 7 d.

**Figure 10 materials-14-03690-f010:**
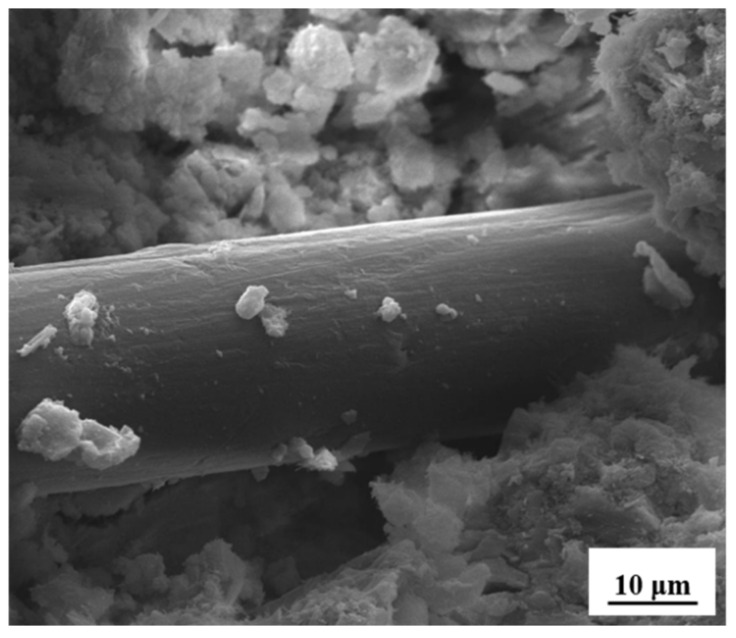
Treated PP fibers are pulled out of mortar.

**Figure 11 materials-14-03690-f011:**
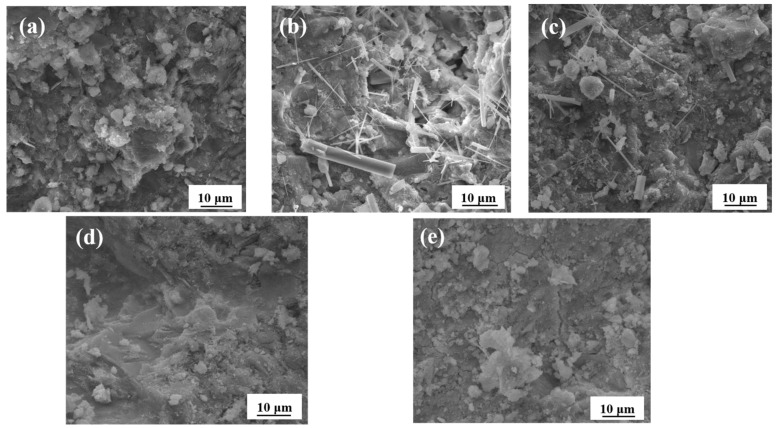
SEM image of treated PP fiber with different dosage after 150 sulfate dry-wet cycles: (**a**) 0%; (**b**) 0.6%; (**c**) 0.9%; (**d**) 1.2%; (**e**) 1.5%.

**Figure 12 materials-14-03690-f012:**
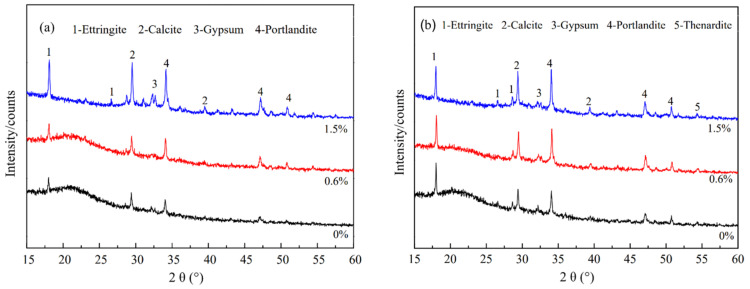
XRD diagram of the hydration product from the same age of the specimen after standard curing and 150 sulfate dry-wet cycles: (**a**) standard curing; (**b**) 150 sulfate dry-wet cycles.

**Table 1 materials-14-03690-t001:** Chemical composition of cement.

Component	CaO	Al_2_O_3_	SO_3_	SiO_2_	Fe_2_O_3_	MgO	Na_2_O	K_2_O	LOI
Content (%)	60.58	3.81	3.63	17.72	3.06	2.95	0.59	0.83	1.43

LOI: Loss on ignition.

**Table 2 materials-14-03690-t002:** Physical properties of ordinary Portland cement.

Fineness (wt.%)	Setting Time (min)		Flexural Strength (MPa)		Compressive Strength (MPa)	
	Initial Setting	Final Setting	3 d	28 d	3 d	28 d
1.8	100	160	4.8	7.9	22.7	47.0

d: day.

**Table 3 materials-14-03690-t003:** The properties of polypropylene fibers.

Fiber Type	Length (mm)	Length-Diameter Ratio	Tensile Strength (MPa)	Elastic Modulus (GPa)	Density (g/cm^3^)
PP fiber	19	200	500	3.5	0.91

**Table 4 materials-14-03690-t004:** Cement mortar mix ratio design.

Number	Cement/Sand Ratio	Cement (g)	Sand (g)	Water (g)	Dosage of Fiber (Vf, %)	Water/Cement Ratio
1	1:1	900	900	405	0.0	0.45
2	1:1	900	900	405	0.6	0.45
3	1:1	900	900	405	0.9	0.45
4	1:1	900	900	405	1.2	0.45
5	1:1	900	900	405	1.5	0.45
6	1:1.5	720	1080	324	0.0	0.45
7	1:1.5	720	1080	324	0.6	0.45
8	1:1.5	720	1080	324	0.9	0.45
9	1:1.5	720	1080	324	1.2	0.45
10	1:1.5	720	1080	324	1.5	0.45
11	1:2	600	1200	270	0.0	0.45
12	1:2	600	1200	270	0.6	0.45
13	1:2	600	1200	270	0.9	0.45
14	1:2	600	1200	270	1.2	0.45
15	1:2	600	1200	270	1.5	0.45

## Data Availability

The data presented in this study are available in insert article.

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
