# Peer review of "Effect of Surface Treatment of Polypropylene (PP) Fiber on the Sulfate Corrosion Resistance of Cement Mortar"

_materials, 2021, doi:10.3390/ma14133690_

Round 1
Reviewer 1 Report
The subjected covered in the submitted manuscript lies within the scope of the journal.
The overall goal of the paper was to investigate the effect that of the surface treatment of polypropylene fibres, with a silane coupling agent, has on the sulfate corrosion resistance of cement mortar. The study comprised the evaluation of mortars with different cement-to-sand ratios and dosages of the treated PP fibre. Furthermore, the study included the assessment of apparent morphology, mass loss rate, flexural strength, corrosion resistance coefficient, as well as the investigation of the mortar microstructure X-ray diffraction and scanning electron microscopy.
The work and results presented in the manuscript are relevant for the scientific community. The paper is well structured and is written in a concise and clear way. The conclusions are well supported by the data presented. However, some amendments are required to improve the readability and correctness of the manuscript:
- A thorough spelling and grammar check shall be performed.
- In the introduction, when authors present a summary on previous studies on the topic, they should also mention which fibre surface treatments were evaluated in each one of the reported studies. Authors shall also mention if there are already studies involving the treatment of fibres with silanes and calcium carbonate dispersions. This is necessary to properly justify the novelty of the research work presented in the manuscript.
- Authors shall refer to Figures consistently throughout the manuscript, currently, authors are using “Figure” and “Fig.” and “Fig”.
- Figure 1 is not mentioned in the text.
- Please correct “Figure 12” to “Figure 13” in page 11.
- In the experimental section, please clarify if the properties presented in Tables 1 and 2 were determined by the authors or given by the manufacturers.
- The experimental section needs to be more detailed. For instance, authors should describe the test methods used for FTIR analysis and contact angle measurements. Please clarify how many specimens were produced for each mortar.
- In section 3.2, please revise wording used in the sentence “The treated PP fiber samples with different C/S ratios and different dosages were eroded by 150 dry-wet cycles of sulfate treatment”.
- In section 3.4, please revise the sentences “The flexural strength and corrosion resistance coefficient of the treated PP fiber mortar test blocks under the erosion of a 5% sodium sulfate solution with different C/S ratios. As shown in Fig 8”.
- In the analyses and discussion section, please include in the graphs the standard deviation of the results to allow for a better assessment of the results obtained.
- Authors shall clarify if the test campaign included mortars produced with untreated fibres. If they did, the results shall be shown. This is essential to assess the benefit gained from the surface treatment.
- Please correct legend of Figure 8.
- The formatting of the citations used throughout the text need to be uniform.
- In the analyses and discussion section, please provide an more in-depth discussion of the results obtained to allow a better perception of the factors studied, for instance include a comparison with results from other studies/authors, etc.; if authors did not find such studies this should be mentioned. Also, when authors attempt to provide an explanation for the observed behaviours, they shall be clear on what can actually be inferred from the tests carried out in the study, and what is merely being proposed/suggested as an explanation. Please revise analysis given in section 3.5, as it does not appear to be entirely coherent with the results shown in Figures 12 and 13.
- Finally, the conclusions need to be updated so that they clear state what were the major novel findings of the study.
Author Response
- A thorough spelling and grammar check shall be performed.
Response 1: It has been checked and modified as required.
- In the introduction, when authors present a summary on previous studies on the topic, they should also mention which fibre surface treatments were evaluated in each one of the reported studies. Authors shall also mention if there are already studies involving the treatment of fibres with silanes and calcium carbonate dispersions. This is necessary to properly justify the novelty of the research work presented in the manuscript.
Response 2: The introduction have been modified. It is modified as follows:
“In addition, owing to the hydrophobic surface (does not absorb water) of the polymer, it does not interfere in the hydration reaction of concrete [18]. However, the PP chain structure is chemically inert and hydrophobic, with a low surface energy. When PP fibers are mixed into the concrete mixture, the fibers form clusters, and uniform distribution cannot be achieved. Clusters of fibers often trap a considerable amount of air, adversely affecting the mechanical properties of fiber-reinforced concrete [19]. Therefore, to extend the application range of the PP fiber concrete, the fiber surface needs to be treated via the increase in the surface roughness of the polymer and introduction of polar groups on its surface, thereby improving the dispersion of the fiber in the concrete [20, 21]. At present, there are many surface treatment methods for PP fibers, such as plasma treatment, surface oxidation and etching, photooxidation surface treatment and radiation grafting treatment, etc. [22-26]. López-Buendía [22] and Lutfur Akand et al. [23] have reported that the surface chemical properties and morphology of chemically treated PP fibers are improved, as well as the adhesion performance between the fibers and concrete, and their surface roughness is increased. The mechanical properties of treated fiber-reinforced concrete composite materials are improved in comparison with that of the untreated fiber concrete. Pietro [24] and Zhang et al. [25] have reported that PP fiber modification can significantly improve the a) uniformity of fiber dispersion in the cement mortar matrix, b) adhesion between the fiber and concrete matrix, and c) the toughness and cracking of the concrete. F. Denes [26] and Li et al. [27] have pointed out that treated PP fiber can improve the mechanical properties of cement-based composites (viz. flexural strength and flexural toughness). Gang [28] and Dai et al. [29, 30] have reported that the incorporation of treated PP fibers can improve the freeze-thaw resistance, impermeability, and erosion resistance of concrete. However, these methods have many shortcomings, such as surface oxidation and etching methods to improve the adhesion at the cost of fiber strength loss, and the chemical reagents used are very harmful to the environment; The equipment continuity degree and stability of plasma surface treatment are poor. The investment of photooxidation surface treatment and radiation grafting treatment equipment is very high, and the optimum process conditions are not easy to master, and the industrial production is difficult. In this paper, nano-sized calcium carbonate and silane coupling agent were used to modify PP fiber.”
- Authors shall refer to Figures consistently throughout the manuscript, currently, authors are using “Figure” and “Fig.” and “Fig”.
Response 3: It has been revised in the manuscript as requested.
- Figure 1 is not mentioned in the text.
Response 4: The description of Figure 1 has been added in 3.1.
“Figure 1 shows the SEM images of the surface of treaded and untreaded PP fibers. As can be seen from Figure 1, the surface of the untreaded PP fiber is very smooth, while the surface of the PP fiber treaded by nano-CaCO3-silane coupling agent is very rough, and the surface of the fiber is attached with nano CaCO3-white fine particles.”
- Please correct “Figure 12” to “Figure 13” in page 11.
Response 5: The "Figure 12" on page 11 has been changed to "Figure 13".
- In the experimental section, please clarify if the properties presented in Tables 1 and 2 were determined by the authors or given by the manufacturers.
Response 6: The properties of Table 1 and Table 2 in the manuscript are provided by the manufacturer.
- The experimental section needs to be more detailed. For instance, authors should describe the test methods used for FTIR analysis and wetting angle measurements. Please clarify how many specimens were produced for each mortar.
Response 7: Experimental methods on FTIR and fiber contact angle have been added in Manuscript 2.4 Analysis Techniques.
The test method of Fourier transform infrared spectroscopy is as follows:
Before the measurement, the fibers were dried in a vacuum oven at 40°C for 2 hours, and the dried fibers were cut into pieces with scissors and compressed into tablets with potassium bromide for testing.
The measuring method of fiber wetting angle is as follows:
Take a 5cm long monofilament fiber and fix it on the support. After the water droplets are stabilized on the fiber surface for 15 seconds, the contact angle between the fiber and water is measured, and the average value is obtained by measuring 3 times. "
Test the strength of specimens after standard curing and sulfate erosion for 30, 60, 90, 120, and 150 times respectively, a total of 150 sets of specimens (refer to the design of cement mortar mix ratio).
- In section 3.2, please revise wording used in the sentence “The treated PP fiber samples with different C/S ratios and different dosages were eroded by 150 dry-wet cycles of sulfate treatment”.
Response 8: Modify the sentence "The treated PP fiber samples with different C/S ratios and different dosages were eroded by 150 dry-wet cycles of sulfate treatment" to "Appearance of PP fiber mortar specimens treated with different C/S and different dosages after dry -wet cycles of 150 times ".
- In section 3.4, please revise the sentences “The flexural strength and corrosion resistance coefficient of the treated PP fiber mortar test samples under the erosion of a 5% sodium sulfate solution with different C/S ratios. As shown in Fig 8”.
Response 9: Modify the sentence“The flexural strength and corrosion resistance coefficient of the treated PP fiber mortar test samples under the erosion of a 5% sodium sulfate solution with different C/S ratios. As shown in Fig 8” to “The corrosion resistance coefficient of flexural strength of the treated PP fiber mortar test specimens under different C/S ratio”This is shown in Figure 8.”
- In the analyses and discussion section, please include in the graphs the standard deviation of the results to allow for a better assessment of the results obtained.
Response 10: β is the mass loss rate of the specimen at a certain age (Precisely keep two decimal places); represents the strength and corrosion resistance coefficient (Precisely keep two decimal places).
- Auhors shall clarify if the test campaign included mortars produced with untreated fibres. If they did, the results shall be shown. This is essential to assess the benefit gained from the surface treatment.
Response 11: The mechanical properties and fluidity of unmodified PP fiber cement mortar were tested. The experimental results show that the submersion test shows that the unmodified PP fiber has poor dispersion and fluidity, and the flexural strength and compressive strength are not as high as the modified PP fiber cement mortar. Therefore, this article mainly studies the effect of modified PP fiber on the sulfate corrosion resistance of cement mortar.
- Please correct legend of Figure 8.
Response 12: The legend in Figure 8 is corrected to “The corrosion resistance coefficient of flexural strength of the treated PP fiber mortar test speci-mens under different C/S ratio (a)1:1; (b) 1:1.5; (c) 1.2”.
- The formatting of the citations used throughout the text need to be uniform.
Response 13: All modifications have been made according to the requirements of the article.
- In the analyses and discussion section, please provide an more in-depth discussion of the results obtained to allow a better perception of the factors studied, for instance include a comparison with results from other studies/authors, etc.; if authors did not find such studies this should be mentioned. Also, when authors attempt to provide an explanation for the observed behaviours, they shall be clear on what can actually be inferred from the tests carried out in the study, and what is merely being proposed/ suggested as an explanation. Please revise analysis given in section 3.5, as it does not appear to be entirely coherent with the results shown in Figures 12 and 13.
Response 14:
The analysis in Section 3.5 is modified as “Figure 12 shows the SEM images of PP fibers with different dosages after 150 cycles of dry-wet sulfate. Unreacted calcium hydroxide was observed in Figure 12(b) and (c) because the mortar was mixed with an appropriate amount of treated PP fibers to prevent the mortar from hardening before the formation of interconnected cracks and holes. Effectively inhibit the invasion of external SO42-. Ettringite crystals were also observed. The expansion of ettringite possibly caused microcracks, but the growth of ettringite around the cracks and the filling of micropores and microcracks helped in increasing the matrix density.It can be seen from Figure 13 that the ettringite peak increases after sulfate attack. As can be observed in Figure 12(a), (d) and (e), ettringite crystals and calcium hydroxide were absent because the fibers were not mixed, and with an extremely high fiber dosage, the matrix was not dense, and connected pores were formed; hence, the etching liquid can enter more easily [38-40]. When the fiber content is 1.5%, a tiny sodium sulfate diffraction peak appears in Figure 13. Therefore, the larger dosage of the fiber and the anhydrous Glauber’s salt with a higher sodium sulfate crystallization pressure produced a higher swelling stress on the pore wall, also considerably accelerating sample damage.”
- Finally, the conclusions need to be updated so that they clear state what were the major novel findings of the study
Response 15: Conclusion is modified as
” 1. The PP fiber surface treated with the silane coupling agent was successfully grafted with hydrophilic groups. The wetting angles of untreated PP fiber and treated PP fiber were 137.2° and 125.9°, respectively. The hydrophilicity of the treated PP fiber was increased by 8.9%.
- The flexural strength and corrosion resistance coefficient of cement mortar with the treated PP fiber were greater than those without PP. When the mortar ratio is 1:1 and the fiber content is 0.6%, the mass loss rate, compressive strength and flexural strength loss of the specimens are all the smallest.
3.The surface-treated PP fiber was immersed for different times in a Na2SO4 solution, and the change in the surface morphology and chemical composition were charac-terized by SEM and EDS. The surface roughness of the surface-treated PP fiber increases and the fiber diameter becomes larger, which improves the interface adhesion between the treated PP fiber and the cement matrix, which is the main reason for improving the sulfate erosion resistance of cement mortar.”

Reviewer 2 Report
The manuscripr deals with the treatment of PP fiber with silane coupling agent before the enhancing of concrete structures. The scope is avoiding the formation of clusters which could determine stress concentrations that may affect the overall mechanical properites. The effects of this treatement on the reinforced concrete were examined with X-ray diffraction and SEM. It was found that the PP fiber formed a proper network within the cement mortar, and that the relative interface adesion was improved.
Overall considered, the manuscript is clear and well written, with adequate English level. Results are suitably presented and conclusions are properly drawn to summurize the main outcomes of the work. Therefore, the manuscript can be accepted for publication in the present form.
Author Response
Dear professor,
Thank you for reviewing my paper in your spare time, and I wish you a happy life.
kind regards
Ma Linlin
Reviewer 3 Report
The article covers the topic of the Effect of Surface Treatment of Polypropylene (PP) Fiber on the Sulfate Corrosion Resistance of Cement Mortar. The manuscript has acceptable cohesion. In my opinion, article presents valuable content.
This is an interesting paper that deals with a timely topic.
Some modification which should be considered to improve the quality of paper:
1. The introduction needs more attention. Please add more literature
related to the essence of this studies. I suggest to explain the effect of the dosage of PP Fibers in cement composites and concrete as a one of key factors determining strength properties.
Some literature could be helpful:The influence of polypropylene fibres on the properties of fresh and hardened concrete, Chajec et al; Effect of Polypropylene Fiber on Properties of Mortar, Salahaldein Alsadey.
2.I suggest to add separated point - Research significance - Please descibe here the main essence of the research. Why presented studies are so important?
3.How was measured tensile strength of PP fibres? Is this data received from manufacturer?
4.Please provide cement content.
5.Please describe figure 2 and add the scale.
6.Which standards were used during determination of flexural strength of specimens?
7. It is recommended to indicate potential application of research results in civil engineering or another discipline.
8.Please add a short introduction to the conclusions such as: The paper describes .....The following conclusions can be stated from the work presented in the article:..
Author Response
- The introduction needs more attention. Please add more literature related to the essence of this studies. I suggest to explain the effect of the dosage of PP Fibers in cement composites and concrete as a one of key factors determining strength properties.Some literature could be helpful:The influence of polypropylene fibres on the properties of fresh and hardened concrete, Chajec et al; Effect of Polypropylene Fiber on Properties of Mortar.
Response 1: I have added more literature related to the essence of this studies as suggested.
Andrei Mustea et al. [15] believe that the dosage of 1,0kg PP fibers per 1m3 mortar represents an optimal value for plaster mortars. ZH Yang ea tl [16] Show that suitable amount of PP fiber is 1. 3 kg/m-3. Under this dosage, PP fiber can not only improve the construction performance of aeolian sand mortar, but also improve its basic mechanical properties. A Chajec et al. [17] reported that the addition of PP fibers to the concrete mixture does not increase the compressive and tensile strength of concrete. Ye J ea tl. reported that adding 1.5% PP fiber can improve the flexural strength of concrete.
I suggest to add separated point - Research significance - Please descibe here the main essence of the research. Why presented studies are so important?
Response 2: The significance and importance of this study have been briefly discussed in the abstract.
Sulfate erosion is one of the most complex and harmful chemical corrosion action. After sulfate erosion, concrete will expand, crack, dissolve, peel off and decrease in strength, which affects the durability of structures. …………The treated fiber has good adhesion to concrete and the fiber is evenly dispersed in the concrete. It can prevent cracks, strengthen and toughen the concrete, and improve the interface performance between the fiber and the concrete.…………However, these methods have many shortcomings, such as surface oxidation and etching methods to improve the adhesion at the cost of fiber strength loss, and the chemical reagents used are very harmful to the environment; The equipment continuity degree and stability of plasma surface treatment are poor. The investment of photooxidation surface treatment and radiation grafting treatment equipment is very high, and the optimum process conditions are not easy to master, and the industrial production is difficult. In this paper, nano-sized calcium carbonate and silane coupling agent were used to modify PP fiber.
How was measured tensile strength of PP fibres? Is this data received from manufacturer?
Response 3:
The stretching of the fiber is the tensile strength of the monofilament and the tensile strength data of PP fiber is provided by the manufacturer.
- Please provide cement content.
Response 4: The chemical composition of cement has been added.
Table 1. Chemical composition of cement.
|
Component |
CaO |
Al2O3 |
SO3 |
SiO2 |
Fe2O3 |
MgO |
Na2O |
K2O |
LOI |
|
Content (%) |
60.58 |
3.81 |
3.63 |
17.72 |
3.06 |
2.95 |
0.59 |
0.83 |
1.43 |
5.Please describe figure 2 and add the scale.
Response 5: Figure 2 has been modified, and the modified results are as follows:
Figure 2 shows wetting angles of untreated and treated PP fibers. The wetting angle of the untreated PP fiber was 137.2°. After treatment, its wetting angle decreased to 125.9°, and the hydrophilicity increased by 8.9%, indicating that the untreated PP fiber exhibits lower wettability. This is because after the composite modification of silane coupling agent-nano calcium carbonate, some hydroxyl groups on the surface of nano calcium carbonate react with sodium polyacrylate to produce hydrophilic carboxyl groups (COO-). Therefore, the fibers exhibited when contacted with water good hydrophilicity.
- Which standards were used during determination of flexural strength of specimens?
Response 6: Determination of flexura strength of specimens according to GB/T 17671-1999 "cement mortar strength test method (ISO method)".
- It is recommended to indicate potential application of research results in civil engineering or another discipline.
Response 7: When fiber is mixed into concrete, it exhibits strain hardening behavior during tension and high toughness during compression. In civil engineering applications, it can play a role in strengthening components and improve the seismic resistance of components.
Please add a short introduction to the conclusions such as: The paper describes .....The following conclusions can be stated from the work presented in the article:
Response 8: A brief introduction has been added to the conclusion section, as shown below:
Aimed at the problems of smooth surface of PP fiber, poor hydrophilicity, poor dispersion of fiber in cement mortar and inadequate bonding of the matrix. In this paper, the surface of PP fiber was treated with silane coupling agent. The effect of treated PP fiber on the sulfate resistance of cement mortar was analyzed by visual observation, the resistance coefficient of flexural strength and microanalysis.

Round 2
Reviewer 3 Report
The majority of remarks have been considered by authors.
The authors responded to comments of the reviewer in good way.
The current version is more satisfactory for reviewer.
Just a minor remark:
Position 16 in literature was incorrectly cited, it should be:
Chajec, A.; Krzywinski, K.; Sadowski, L.; Ostrowski, K. The influence of polypropylene fibres on the properties of fresh and
hardened concrete. Technical Transaction 2019, 5, 71–82.
Considering the above, I suggest that article could be published.